

# Metabolic tradeoffs and heterogeneity in microbial responses to temperature determine the fate of litter carbon in a warmer world

Grace Pold[1*], Seeta A. Sistla[2], and Kristen M. DeAngelis[3]

[1]Graduate Program in Organismic and Evolutionary Biology, University of Massachusetts, Amherst, MA 01003, USA
[2]Natural Resources Management and Environmental Sciences, California Polytechnic State University, San Luis Obispo, CA 93407, USA
[3]Department of Microbiology, University of Massachusetts, Amherst, MA 01003, USA

**Correspondence:** Grace Pold (apold@umass.edu)





**Abstract.** Climate change has the potential to destabilize the Earth's massive terrestrial carbon (C) stocks, but the degree to which models project this destabilization to occur depends on the kinds and complexities of microbial processes they simulate. Of particular note is carbon use efficiency (CUE), which determines the fraction of C processed by microbes that is anabolized into microbial biomass rather than being lost to the atmosphere as carbon dioxide. The temperature sensitivity of CUE is often modeled as a homogeneous property of the community, which contrasts with empirical data and has unknown impacts on projected changes to the soil carbon cycle under global warming. We used the DEMENT model—which simulates taxon-level litter decomposition dynamics—to explore the effects of introducing organism-level heterogeneity into the CUE response to temperature for decomposition of leaf litter under 5°C of warming. We found that allowing CUE temperature response to differ between taxa facilitated increased loss of litter C, unless fungal taxa were specifically restricted to decreasing CUE with temperature. Increased loss of litter C was observed when the growth of a larger microbial biomass pool was fueled by higher community-level average CUE at higher temperature in the heterogeneous microbial community, with effectively lower costs for extracellular enzyme production. Together these results implicate a role for diversity of taxon-level CUE responses in driving the fate of litter C in a warmer world.





## 1 Introduction

Soil heterotrophs are central to the cycling and recycling of the 60 Gigatons of organic carbon (C) that plants deposit onto and into the ground each year. How well these litter inputs are converted into relatively stable soil organic matter depends on temperature, moisture, chemical composition, and soil mineralogy, which interact to influence microbial physiology (Manzoni et al., 2012; Kallenbach et al., 2016; Oldfield et al., 2018). Predictions regarding how soil C stocks will respond to climate change are, in turn, highly sensitive to how carbon use efficiency (CUE)—or the fraction of C taken up by a cell and incorporated into biomass rather than being respired—-changes with temperature (Allison et al., 2010; Wieder et al., 2013; Allison, 2014; Li et al., 2014; Sistla et al., 2014; Tang and Riley, 2015). As such, quantifying microbial decomposer CUE and its responsiveness to environmental change has been subject to intensive study (Devêvre and Horwáth, 2000; Frey et al., 2013; Blagodatskaya et al., 2014; Lee and Schmidt, 2014; Spohn et al., 2016a, b; Öquist et al., 2017; Malik et al., 2018; Geyer et al., 2019; Malik et al., 2019; Zheng et al., 2019).

Soil microbial communities show considerable differences in how their metabolisms respond to elevated temperatures, with their CUE increasing (Öquist et al., 2017; Zheng et al., 2019), decreasing (Devêvre and Horwáth, 2000; Frey et al., 2013; Öquist et al., 2017; Li et al., 2018; Zheng et al., 2019) or remaining unaffected by warming (Dijkstra et al., 2011b; Öquist et al., 2017; Walker et al., 2018; Zheng et al., 2019). However, models of the soil C cycle generally assume either no change (Allison et al., 2010; Li et al., 2014; Wieder et al., 2014) or a fixed decrease in CUE with temperature (Allison et al., 2010; Wieder et al., 2013; Allison, 2014; Li et al., 2014). When CUE is allowed to directly increase with temperature, this temperature response is fixed across taxa (Frey et al., 2013; Ye et al., 2019). In other instances, CUE may be modeled as fixed within taxa, such that changes in community-level CUE with warming are the result of shifts in the dominant group or groups of organisms present as a function of their dietary preferences and/or C:N ratio (Wieder et al., 2013; Sistla et al., 2014). Therefore, models have thus far insufficiently accounted for how the temperature sensitivity of central metabolism may differ between microbes, such that intrinsic differences in efficiency between taxa above and beyond temperature-driven differences in substrate supply may also drive microbial community trajectories.

Variation in the temperature sensitivity of growth efficiency could be driven by differences in the rate-limiting step of central metabolic pathways (Dijkstra et al., 2011a), or in how well the proteins responsible for the extracellular processing and uptake of environmental nutrients are able to maintain activity as temperature increases (Allison et al., 2018; Alster et al., 2018). For instance, there is some evidence that bacteria benefit more than fungi from an increase in temperature, as their growth rate was observed to decrease less rapidly with temperature above its optimum than the fungal community's did (Pietikäinen et al., 2005). Although the respiration rate for the two groups could not be isolated in that study, it is possible that they may also differ in the temperature response of CUE as a consequence of changing nutrient demands (Keiblinger et al., 2010; Sinsabaugh et al., 2016). Therefore, the temperature range over which an an organism can maintain efficient growth is one important dimension of its niche (Cavicchioli, 2016), and, when combined with other "response traits" determining how they react to the environment (Lavorel and Garnier, 2002), can impact their success in the environment. These response traits may in turn be linked to "effect traits" determining how an organism alters its environment, such as extracellular enzyme production (Martiny et al., 2013;





Treseder and Lennon, 2015; Amend et al., 2016). For instance, taxa capable of growth at higher temperatures may need to

produce a broader suite of enzymes and attack a wider range of substrates to support this rapid growth than slower-growing taxa.

Extracellular enzyme production is proposed to impose substantial metabolic costs on the cell, however. This is because carbon which could otherwise be allocated to growing the cell or generating the energy required to maintain it must instead be spent producing amino acids and expending ATP to link them together (Kaleta et al., 2013; Kafri et al., 2016). As such,

extracellular enzyme production is inferred to reduce the carbon use efficiency of soil microbial communities (Allison, 2014; Malik et al., 2019).

We explored whether interactions between the temperature sensitivity of intracellular (i.e. CUE) and extracellular (i.e. litter decomposing enzyme) metabolic processes of cells can explain why CUE is observed to increase with temperature in some soils, and decrease in others. We used the litter decomposition model DEMENT (Allison, 2012) to evaluate four hypothe-

ses: 1) allowing temperature response of CUE to vary between taxa increases uncertainty in projected litter decomposition dynamics because more diverse phenotypic combinations exist for competitive selection (i.e. species sorting) to act upon; 2) this variation favors a community with higher CUE, in turn leading to higher microbial biomass and greater litter C loss with warming; 3) forcing the temperature response of CUE to positively co-vary with the number of enzymes an organism produces causes greater litter C loss than when the two factors vary independently, because increasing CUE with temperature offsets

the increased costs against CUE associated with copious enzyme production; and 4) the magnitude of litter carbon loss with warming is greater when the carbon-rich fungal functional group increases with warming than if only the nitrogen-rich bacterial functional group does.

## 2   Methods

### 2.1   DEMENT Background and Model Design

DEMENT (Allison, 2012) is a litter decomposition model designed to simulate the loss of leaf C through time. The principal advancement of DEMENT over its predecessors is that it is both microbially- and spatially-explicit. The model is able to simulate inter- and intraspecific microbial interactions, with a primary focus on the tradeoff between the ability to take up and digest substrates, and the metabolic costs of creating and maintaining the machinery required to do so. Because these tradeoffs are both explicit and variable across taxa, DEMENT is an ideal model for evaluating how the physiology and ecology of

microbes affects C stocks in a changing world. Furthermore, DEMENT allows for consideration of how diversity in responses across taxa (rather than using some cross-taxon mean) can facilitate soil C responses to climate change. Full details about the setup and execution of DEMENT are available elsewhere (Allison, 2012, 2014; Allison and Goulden, 2017); here we describe the model controls on CUE which are relevant to our study.

Intrinsic CUE—the maximum CUE an organism could attain under ideal temperature and stoichiometry—is calculated for

each taxon as a function of the baseline CUE at 15 °C, and the number of enzymes and transporters the taxon can produce. In turn, how much CUE is decreased due to enzyme and transporter production depends on the cost per enzyme ($C_e$) and cost per





transporter ($C_u$) (Table 1). The C used in enzyme synthesis is considered a loss from the cell, and is therefore not reported as microbial biomass C. The intrinsic CUE of each taxon is adjusted for temperature, decreasing by $0.016^{\circ}C^{-1}$ by default (i.e. $C_t$ = $-0.016^{\circ}C^{-1}$), consistent with a global meta-analysis (Qiao et al., 2019).

## 2.2  Modifications to DEMENT

Baseline CUE ($C_r$) was adjusted downwards from its original published value of 0.58 to 0.38 at $15^{\circ}C$; this not only improved model stability (Table S1), but is also consistent with a comparative modeling study completed by Li et al. (2014), several $^{18}O$-$H_2O$ based CUE measurements (Spohn et al., 2016a, b; Geyer et al., 2019), and for the structural components of litter modeled by MIMICS (Wieder et al., 2015a). We also altered the temperature sensitivity of CUE ($C_t$) from its default (fixed at a given cross-taxon average), to vary around the mean in different ways (Figure 1). In the first set of scenarios, $C_t$ varied independent of the taxonomic identity or number of enzymes a taxon produced. In the second scenario, $C_t$ was limited to either increasing or decreasing as a function of the number of enzymes a given taxon had. In the third, bacteria and fungi were constrained to both have a positive $C_t$, both a negative $C_t$, or one a positive and the other a negative $C_t$. In all instances, $C_t$ was selected at random from a uniform distribution bounded by +/- $0.022^{\circ}C^{-1}$ at the upper and/or lower limits. These values are within the range of temperature sensitivities observed for both bacterial cultures in the lab and for field communities (Figure 2), as well as values inferred based on modeling CUE against mean annual temperature on a global basis (Sinsabaugh et al., 2017; Ye et al., 2019). It was necessary to force the temperature sensitivity of CUE to take on a zero-centered uniform distribution so that simulation outputs in which extracellular enzyme counts were linked to the $C_t$ could be compared to those scenarios where they were not linked, without changing the distribution of extracellular enzyme counts present in the community.

## 2.3  Running DEMENT

DEMENT v0.7.2 was downloaded from GitHub, and modified as described above. DEMENT was subsequently run on the Massachusetts Green High Performance Computing Cluster for 6,000 model days using 59 different independent starting seeds and a 100x100 grid size. "Control" runs were completed at $15^{\circ}C$ (equivalent to April to November mean soil temperature for a northern mid-latitude temperate deciduous forest (Boose, 2001)), while "heated" runs were completed at $20^{\circ}C$ (Allison, 2014). The first 1000 days of each resultant output file was excluded from the analysis because of rapid shifts in the microbial community during this time. In addition, outputs were filtered to exclude any seeds where the substrate pool was two or more times greater at the end of the model run than the median during the preceding 5000 days, indicating unrealistic, unconstrained litter accumulation. R version 3.4.0 was used for all runs and analyses (R Core Team, 2016). A full set of parameters and the model used to run all these simulations can be found in the supplementary files "params.txt" and "DEMENTmodel.R", respectively.



## 2.4 Analysis of Outputs

The model outputs of interest were litter organic matter (LOM), microbial biomass carbon (MBC), respiration rate, richness and diversity of the surviving community, median number of enzymes per taxon for taxa alive during the 5000-day simulation, fungal:bacterial biomass ratio for surviving taxa, and biomass-weighted CUE at 15 and 20ºC. Richness was calculated as the

number of taxa surviving to the end of the 5000 day run, while diversity was calculated as the median daily Shannon's H for the duration of the simulation using the vegan package (Oksanen et al., 2017). In order to determine whether warming and model parameterization affected model outputs, we used mixed effect models with starting seed as a random effect and warming or simulation scenario as fixed effects using lmer in lme4 v 1.1-17 (Bates et al., 2015). Data were visually assessed for normality and homoskedasticity using qqplots and residual plots following log-transformation. Significantly different pairwise differences

were subsequently identified using emmeans v.1.3.0 (Lenth et al., 2019), with a stringent Bonferoni-corrected p-value cutoff of $P < 0.0001$. Warming effect sizes are plotted as the natural log ratio of model outputs in heated:control scenarios.

## 3 Results and discussion

LOM and MBC content were both generally higher than observed in environmental samples, leading to MBC:LOM ratios at the high end of ranges observed in the field (2-11% vs. 1-5% (Santos et al., 2012; Xu et al., 2013)). LOM and MBC values

were within the range previously observed for simulations using DEMENT with daily litter inputs (Allison et al., 2014), but greater than those with just a single litter pulse (Allison, 2012; Allison and Goulden, 2017; Evans et al., 2017), indicating that these high biomass and litter C values can be attributed to these substrate inputs.

### 3.1 Intertaxon variability

To evaluate the effect of intertaxon CUE variablity on LOM stocks, we ran the model at 20ºC ("heated") under two scenarios,

and then compared the results to runs at 15ºC ("control"). In the first "heterogeneous" scenario, $C_t$ was assigned from a random uniform distribution bounded by -0.022 and 0.022ºC$^{-1}$ (Figure 1A). In the second scenario, all taxa had an identical temperature sensitivity that was equivalent to the cross-taxon mean (0ºC$^{-1}$) of the starting community in the first scenario (Figure 1B).

Introducing intertaxon differences in CUE temperature response caused the characteristics of the initial microbial community (starting seed) to have a greater impact on litter decomposition than when all taxa had an identical temperature response

(Table 2). This contrasts with the dampening effect proposed to explain instability in small-scale microbially-explicit models compared to their macroscale counterparts (Wieder et al., 2015b), whereby the additive effect of increased physiological diversity was to increase, rather than decrease, uncertainty in the present simulations. The median-standardized interquartile ranges of both MBC (0.25 vs. 0.15) and LOM (0.28 vs. 0.14) increased with the introduction of a variable $C_t$. Through species sorting, this heterogeneously-responding microbial community became more uneven with warming, with similar richness but

lower diversity than the control community (Table 2). The heterogeneous communities maintained a higher median microbial biomass—driving two and a half times more LOM loss—than the homogeneous communities (Figure 3). Intriguingly, neither





litter (r=0.16, p=0.23) nor microbial biomass pool sizes (r=-0.44, $P < 0.001$) positively correlated with extracellular enzyme investment; thus, a (non-significant) 28% increase in the median enzyme count is unlikely to have driven the increased decomposition under the heterogeneous scenario. Instead, increased decomposition and increased biomass are likely the consequence

of elevated CUE under warming conditions.

The homogeneous community scenario tested here is akin to the "no adaptation of CUE" scenario reported in a number of other studies (Allison et al., 2010; Li et al., 2014; Sistla et al., 2014), because the cross-taxon mean used is zero temperature response. Our results of reduced LOM loss in the absence of acclimation are consistent with two previous studies, but contrast with others. In an ecosystem-level model parameterized for an arctic tundra system, Sistla *et al.* (2014) found that greater soil

organic matter (SOM) loss occurred with warming when the microbial community was able to acclimate its CN ratio (and in turn efficiency), than when the CUE was effectively fixed. Likewise, Allison (2014) found greater potential for increased LOM accumulation under warming when there was greater absolute variation in CUE across taxa (i.e. $C_r$ from 0.18 to 0.58 rather than 0.38 to 0.58) (Allison, 2014). On the other hand, a comparison of models where the microbial community is modeled homogeneously showed that soil organic matter loss increases when organisms do not adapt (Li et al., 2014). Similarly, Wieder

*et al.* found that greater SOM loss occurred if the CUE was directly insensitive to temperature than when $C_t$ was negative (Wieder et al., 2014). These microbially-explicit decomposition models vary in if and how they link CUE to microbial traits, and so our findings support the concept that nuances in how different components of CUE respond to warming is an important control on the fate of litter C (Hagerty et al., 2018).

### 3.2    Confirming the role of $C_t$ as an additional niche dimension

We allowed for CUE to increase with temperature for a subset of taxa in a way that most previous modeling efforts have not, and so it is possible that our results deviate from those of prior studies not because of variation in $C_t$, but rather because our simulations explore novel (*P*ositive $C_t$) parameter space. To facilitate comparison with previous decomposition modeling studies, we ran DEMENT simulations to test the effect of $C_t$ being homogeneous vs. heterogeneous when CUE was either always positive (homogeneous $C_t = 0.011$ °C$^{-1}$ (Figure 1A$_i$), heterogeneous $C_t = 0$ to 0.022 °C$^{-1}$ (Figure 1G)) or always

negative (homogeneous $C_t = -0.011$ °C$^{-1}$ (Figure 1A$_{ii}$), heterogeneous $C_t = -0.022$ to 0 °C$^{-1}$ (Figure 1H). In contrast to when $C_t$ was allowed to vary over the whole spectrum of values, introducing heterogeneity in CUE did not increase inter-run uncertainty in LOM or MBC pools (Table S2). We also found that less LOM accumulated when CUE showed a variable decrease with warming than a fixed one (Figure S1), which could be attributed to a reduction in MBC. By contrast, the homogeneous zero-centered and homogeneous positive $C_t$ scenarios, and the heterogeneous zero-centered and heterogeneous positive $C_t$, behaved

more similarly to one-another in that warming decreased LOM while increasing MBC and CUE to a greater degree in the heterogeneous than homogeneous scenarios (Table S2). This finding reinforces the idea that if warming favors decomposer taxa capable of maintaining efficient growth, then soil C loss will be accelerated. Nonetheless, the strongly selected-for positive CUE response is rarely observed in complex soil communities. This indicates that additional tradeoffs with CUE temperature response are likely at play when CUE is either unaffected or decreases with temperature, but that these tradeoffs are missing in





the formulation of DEMENT used in this scenario. One such tradeoff possible to explore within the framework of DEMENT
is the allocation of resources to extracellular enzyme activity.

### 3.3  Linkages between CUE temperature response and extracellular enzyme allocation

Microbes depend upon extracellular enzymes to break down substrates in the environment into digestible pieces, and enzyme
activities are, like CUE, responsive to temperature (Wallenstein et al., 2010; German et al., 2012; Allison et al., 2018). Soil

extracellular enzymes often are active *in-situ* at temperatures much below their activity optima (German et al., 2012; Pold
et al., 2017; Alster et al., 2018). Therefore, warming enables them to process substrates at a higher rate, increasing the supply
of growth substrates to microbes. However, the affinity of enzymes for their substrates also decreases as temperature increases
(German et al., 2012; Allison et al., 2018); unless enzyme $V_{max}$ increases faster with temperature than $K_m$, additional resources
must be diverted from growth to enzyme production to maintain microbial growth substrate supply rate. Therefore, taxa may

differentially-allocate resources to enzymes and so demonstrate a relationship between the temperature sensitivity of CUE and
the number of enzymes they produce.

We evaluated whether litter decomposition changed its trajectory when the organisms with the greatest genomic potential to
break the litter down (i.e. enzyme counts) also showed the most- or least-positive growth efficiency response to warming. In the
"increase" scenario, we simulated a positive relationship between temperature sensitivity of CUE and extracellular enzymes,

where $C_t$ increased linearly from -0.022°C$^{-1}$ for organisms with no extracellular enzyme production potential to 0.022°C$^{-1}$ for
those organisms capable of producing the model maximum of 40 enzymes (Figure 1C). In the "decrease" scenario, we simulated
a negative relationship between temperature sensitivity of CUE and extracellular enzymes, where the opposite relationship was
imposed with $C_t$ decreasing with enzyme counts (Figure 1D). These scenarios were then compared to the "heterogeneous"
scenario (aka "no relation", as described above), where $C_t$ varied across the same range, but independently of the number of

enzymes an organism could produce. Therefore, the starting distribution of $C_t$ and enzymes per taxon was identical across
scenarios, and only their relationship with one-another changed.

More taxa survived to the end of the simulation when warming was applied under the "increase" scenario than either the
"decrease" or "no relation" scenario (median of 13 versus 7 and 8, respectively, median absolute deviation = 2.97 in all cases).
Under the "increase" scenario, taxa had 70% more enzymes each than the "no relation" scenario, and more than three times

as much as the "decrease" scenario (Table 2). This relationship caused the CUE of surviving taxa to be 20-37% lower at 15°C
for the "increase" scenario compared to the others, but this deficit was diminished at 20°C. As a result, the "increase" scenario
led to higher respiration and a greater LOM loss under warming than under the "decrease" scenario, despite an overall smaller
microbial biomass pool (Figure 4).

How the relationship between $C_t$ and extracellular enzymes drives favorable trait combinations can also be observed in Fig-

ure 5. Surviving taxa retained a median enzyme count of at least 30 and a realized CUE temperature response of no less than
0.0158°C$^{-1}$ under the "increasing" scenario ($\rho$=0.64, $P < 0.001$), but there was no relationship between realized CUE tem-
perature response and enzyme production under either the "decrease" or "no relation" scenarios. The selection for community
capable of maintaining high CUE at high temperatures was much weaker when it was associated with reduced enzyme produc-





tion. When there was no relationship between $C_t$ and extracellular enzyme production costs, however, communities were able

to attain a high realized CUE temperature response over a much wider range of median enzyme costs.

These findings indicate that response traits—which determine how an organism reacts to changes in temperature (e.g. CUE temperature response)—and effect traits—which determine how an organism alters its environment (e.g. litter decomposition potential)—interact to determine the fate of organic C. However, contrary to our hypothesis, adjusting DEMENT to allow for this tradeoff did not substantially alter how community-level CUE responds to temperature. The observation that LOM

is reduced further when enzyme production is effectively cheaper contrasts with earlier work with DEMENT (Allison, 2014) showing smaller litter C pools under both ambient and elevated temperature when enzymes and transporters were cheaper to produce. However, our results are consistent in that microbial biomass was lower when enzyme costs are high, and that the microbial community was able to maintain a higher CUE under warming no matter the enzyme costs. The mechanisms underlying these phenomenological similarities differ, however, due to differences in how $C_t$ was parameterized in the two sets

of model simulations. Specifically, although microbes were able to attain high CUE at elevated temperatures in our simulations by balancing the benefits of elevated CUE at higher temperatures with the costs of enzyme production against CUE, CUE always decreased with temperature in earlier work with DEMENT (Allison, 2014). Furthermore, enzyme production costs varied both with and independently of enzyme counts in previous DEMENT simulations (Allison, 2014).

Increased CUE is likely needed to offset the costs of extracellular enzyme production if taxa are to remain competitive at

elevated temperatures within the framework of DEMENT, but empirical support for correlations between temperature sensitivity of CUE and enzyme investment are needed to best put these inferences to use. By examining correlations between the number of enzymes an organism can produce and its CUE temperature response at the end of the DEMENT model run, we see that there is likely to be either no or a positive correlation between the two variables, rather than a negative one (Figure 6). Limited data from bacterial isolates grown in the lab also support this, whereby CUE temperature response is either positively

correlated or uncorrelated with the number of enzymes produced (Pold et al., in prep). Furthermore, although the mechanisms underlying the isolate response remain unclear, it is consistent with the scenario of $C_t$ and enzyme counts being positively correlated. Specifically, we found isolates with lower CUE at 15°C (more extracellular enzymes in DEMENT) were more likely to have a positive CUE temperature response than those with a higher CUE (fewer enzymes in DEMENT). Together, these insights support a synergism between CUE temperature response and enzyme production, rather than a tradeoff. Because our

DEMENT simulations indicate that selection for organisms characterized by high, positive CUE temperature responses with warming can alter both the directionality and extent of projected C loss, we propose it is important for other models to explore how possible increases—rather than just decreases—in $C_t$ affect terrestrial C projections.

### 3.4 Ecological relevance of microbial metabolic diversity—bacteria vs. fungi

Across scenarios, fungi generally dominated the microbial biomass C pool (Table 2), as is typical for litter decomposition

(Chapman et al. (2013), and references therein). This pattern occurred despite generally lower biomass-weighted CUE for surviving fungal taxa, and preferential loss of fungal taxa across most scenarios. The lower CUE for surviving fungi was not driven by higher enzyme costs than for bacteria, as median biomass-weighed enzyme costs were not statistically different (*P*





> 0.4) and differed by less than one enzyme for the two groups. To test whether modeled differences in fungal vs. bacterial cell sizes and stoichiometry were driving this pattern, we tested how forcing differences in $C_t$ in the two groups would impact
the decomposition rate.

DEMENT was run with CUE simulated to respond: 1) negatively to temperature for all fungi and positively for all bacteria (F-B+; Figure 1E); 2) negatively for all bacteria and positively for all fungi (F+B-; Figure 1F); 3) positively for all bacteria and fungi (F+B+; Figure 1G); or 4) negatively for all bacteria and fungi (F-B-; Figure 1H). Minimum and maximum $C_t$ were set to -0.022 °C and 0.022 °C, respectively.

Litter C accumulated at higher rates with warming when fungal $C_t$ was negative, regardless of the bacterial $C_t$ (Figure 7, blue, orange). This is consistent with the observation that the CUE of surviving fungi was lower at the simulation temperature in seven out of the eleven scenarios. Because fungi have higher CNP ratios than bacteria (and thus higher C demands per unit biomass), we predicted that if fungi have a negative CUE temperature response, they would be weaker competitors at higher temperature than bacterial taxa, reducing their C demand and mitigating the warming effect on SOC stocks. While this
contrasts with the premise that fungi should have a higher CUE (Six et al., 2006) due to their higher CN ratio (Zak et al., 1996), it is consistent with a growing body of literature indicating that substrate quality—rather than the F:B ratio correlated with it—is the underlying driver of differences in CUE between soils (Frey et al., 2013; Thiet et al., 2006; Malik et al., 2018; Soares and Rousk, 2019). Despite the low nutrient content of the daily inputs to the model (92:0.26:0.02 C:N:P), microbes did not show evidence for nutrient limitation as biomass CN and CP ratios were lower than are typical for soil communities (Xu
et al., 2013) (4.1 and 36.7 vs. 7.6 and 42.4, respectively). Thus C was limiting, which could have further disfavored the highly C-demanding fungi when their $C_t$ was negative. The lower CUE (and increased sensitivity to warming) for fungi compared to bacteria under a given scenario was also not driven by increased metabolic costs for enzyme production in fungi, as median biomass-weighted enzyme counts were statistically indistinguishable from those in bacteria.

The litter C pool decreased when fungal CUE increased with temperature (Figure 7), correlating with a smaller microbial
biomass pool when bacterial CUE also decreased with temperature. By contrast, the LOM and MBC responses to warming were similar when only the $C_t$ of bacteria was changed, indicating that it is the fungal warming response which really drives changes in litter decomposition in DEMENT. This result is interesting because no *a priori* differences in decomposition or uptake potential were imposed on the two groups, and fungal and bacterial richness was initially equivalent. Warming decreased the enzyme costs when fungal $C_t$ was positive but bacterial $C_t$ was negative, and decreased them under the opposing scenario, as
evidenced by an increase in $C_r$ in the former and decrease in the latter. Nonetheless, as long as both bacterial and fungal CUE did not both decrease their CUE with temperature, community level CUE remained higher at 20°C than it was at 15°C.

Empirical evidence for high-level differences in the temperature sensitivity of CUE in bacteria and fungi is currently mixed, but indicate CUE temperature response for fungi is unlikely to be more positive than that for bacteria. Zheng et al. (2019) did not find a correlation between the lipid-based fungal:bacterial ratio and $Q_{10}$ of CUE over a range of soils. However, we
(Pold et al., in prep) and our colleagues (Eric Morrison, personal comment) have found that fungi tend to show a stronger negative CUE response with warming than do bacteria when examining them in isolation in the lab. This is consistent with the observation that fungal CUE decreases more strongly with warming than bacterial CUE does when $C_t$ is restricted to negative





values (Figure 7). It is also consistent with the premise that bacterial growth benefits more from elevated temperature than does fungal growth in some soils (Pietikäinen et al., 2005). Greater empirical insight into the taxonomic drivers of the temperature

sensitivity of CUE will assist with constraining the parameterization and projections of microbially-explicit decomposition models such as DEMENT.

### 3.5 Comparison to empirical warming studies

Litter decomposition is typically observed to accelerate under warming (Lu et al., 2013). However, both the chemical composition of the litter and the identity of the living plant community at the site of decomposition are important for the magnitude

of this response (Cornelissen et al., 2007; Ward et al., 2015). Consistent with these empirical studies—but inconsistent with a previous publication using DEMENT (Allison, 2014)—we found that litter decomposition was accelerated by warming in seven of ten scenarios. The range of losses and gains of litter C we observed with warming (-62% (scenario C) to +42% (scenario $A_i$)) approximates the -65% to +36% change observed in field experiments (Lu et al., 2013), with the upper limit only being exceeded when $C_t$ is constrained to negative values. Likewise, values for simulated litter respiration response (-5 to +6%)

fell within those observed for soil respiration in the field (-48 to +178%), and responses for microbial biomass C were also within the observed range (-25 to +58% vs -47 to +86%) (Lu et al., 2013). Our modeled responses to warming thus suggest that one possible explanation for differences in terrestrial C pool responses to warming may be diverse temperature sensitivities of underlying decomposer communities. Nonetheless, a number of additional factors must be taken into consideration when interpreting our results in the context of global climate change, including soil mineral-mediated modulation of substrate supply

(Schimel and Schaeffer, 2012; Coward et al., 2018), plant-microbe feedbacks (Melillo et al., 2011; Sistla et al., 2014; Suseela and Tharayil, 2018), and temporal variation in temperature.

While we explored how different relationships between temperature sensitivity of CUE and enzyme count or taxonomic affiliation affects the C cycle, empirical support for these scenarios is lacking. By including additional dimensions to the warming response we showed that species effectively sort to make communities with differing impacts on litter decomposition.

Following up on the direction and magnitude of these underlying metabolic scenarios in the soil and litter can help us better constrain our model results on reality. New and emerging work indicates that the temperature sensitivity of CUE is weakly negatively correlated with the temperature sensitivity of extracellular enzyme activity, and with bacterial—but not fungal—biomass on the community level (Zheng et al., 2019). However, different kinds of bacteria and fungi are able to show positive, negative, or no temperature sensitivity of CUE, and so high-level assumptions about how these groups respond are unlikely to

resolve uncertainties about the magnitude of soil C loss under warming.

### 4 Conclusions

Our results indicate that accounting for heterogeneous temperature response increases uncertainty regarding future litter C stocks, but only when $C_t$ does not differ from zero on average. However, by combining simulations, empirical studies, and literature searches, we can conclude that microbes with high enzyme costs are likely to have larger increases in intrinsic CUE
with temperature; that taxa can sort on a CUE temperature response axis; and that fungi are more likely to increase CUE with warming than bacteria. The simulations meeting each or all of these criteria lead to loss of litter C under warming, indicating that litter is likely to become a net atmospheric C source in a warmer world. We encourage models functioning on larger scales to explore the effect of including heterogeneity in the temperature response of CUE in order to determine the robustness of our conclusions to other model structures. However, ultimately increased integration of the growing body of literature on the

temperature sensitivity of CUE must be explored for root causes of heterogeneity in temperature sensitivity of CUE in taxa under *in situ* conditions.

*Code and data availability.* The data and modified version of DEMENT used to generate it are available at OSF (https://osf.io/cwep9/?view_only=de0e6ba7b7da493f96531f398ca62c2c) under DOI 10.17605/OSF.IO/CWEP9

*Author contributions.* GP SAS and KMD conceived the experiment; GP completed the experiment and analyzed data; GP wrote the first

draft of the manuscript; all authors contributed to revising the manuscript and interpreting the results

*Competing interests.* The authors declare no competing interests

*Acknowledgements.* The authors wish to thank Steve Allison for sharing DEMENT and providing insight into troubleshooting the model. Funding for this project came from the US Department of Energy grant DE-SC0016590 to KMD and SAS, and from an American Association of University Women American Fellowship to GP.



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

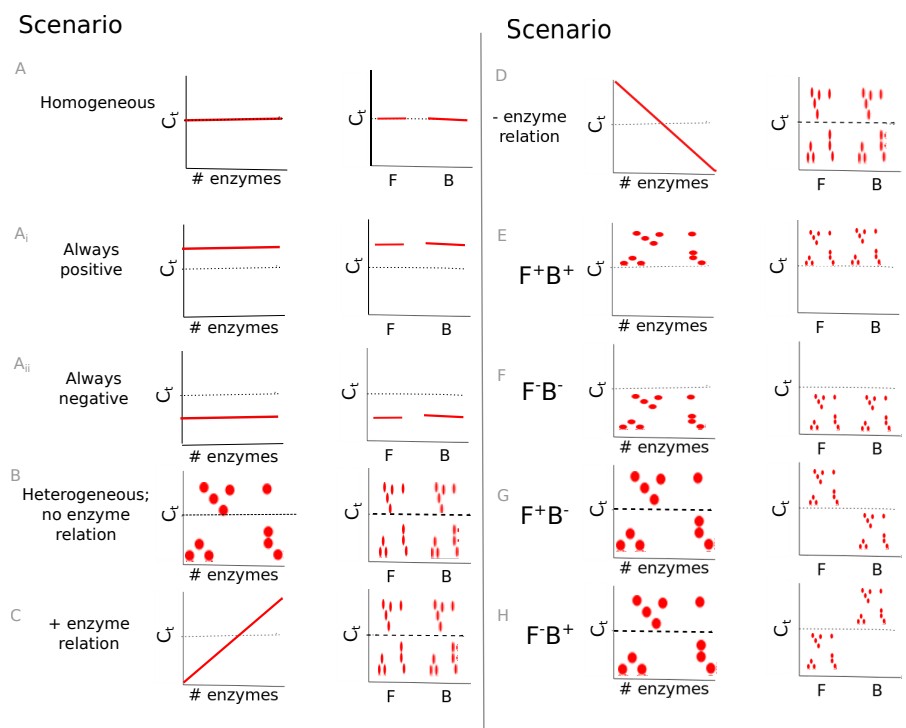

**Figure 1.** Schematic of experimental design used in this study, where CUE temperature response ($C_t$) varies as a function of the number of enzymes and/or taxonomic affiliation of organisms. Graphs show the effects of having homogeneous (A) or heterogeneous (B) $C_t$ across taxa; the effect of forcing a positive (C) or negative (D) correlation between the number of enzymes and $C_t$; effect of fungi and bacteria both having increases (G) or decreases (H) in CUE with temperature, or with one group showing an increase while the other decreases its CUE with temperature (E,F). Horizontal dashed lines indicate a $C_t$ of zero, and clusters of points above and below this line denote when CUE tends to increase or decrease with increasing temperature. The letters F and B in the x-axis of individual graphs denote sensitivities for fungi and bacteria, respectively.


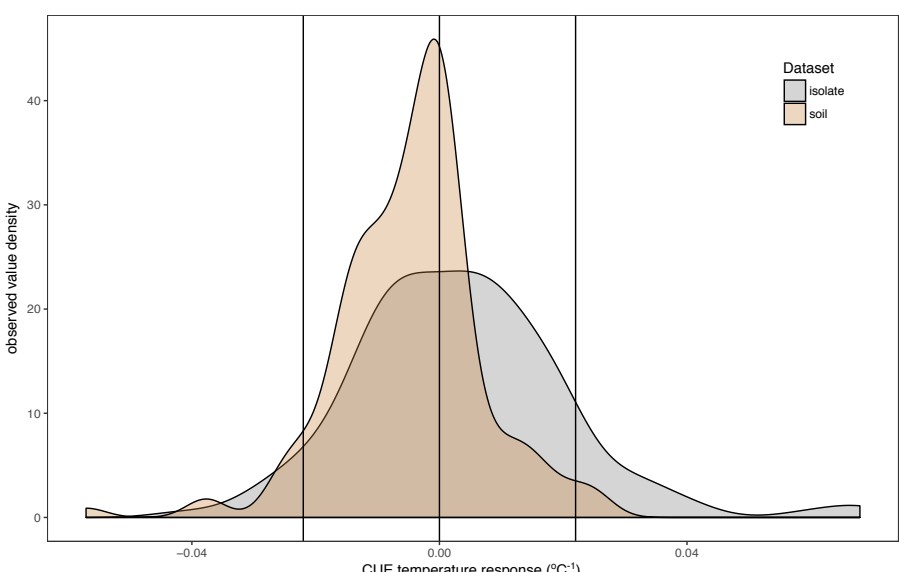

**Figure 2.** Density plot of observed CUE temperature response for 23 soil bacterial isolates grown between 15 and 25°C on four different liquid media types in the lab (n=160 datapoints, grey), and for soil microbial communities grown with various different substrates and temperatures based on a literature search (n=141 datapoints, brown). Vertical lines are placed at 0 (no change in CUE with temperature) as well as the +/-0.022 °C$^{-1}$ upper and lower limits used in the present study. Contributing datapoints are primarily derived from Qiao et al. (2019) and can be found in the "Ct_literatureValues.txt" file in OSF.


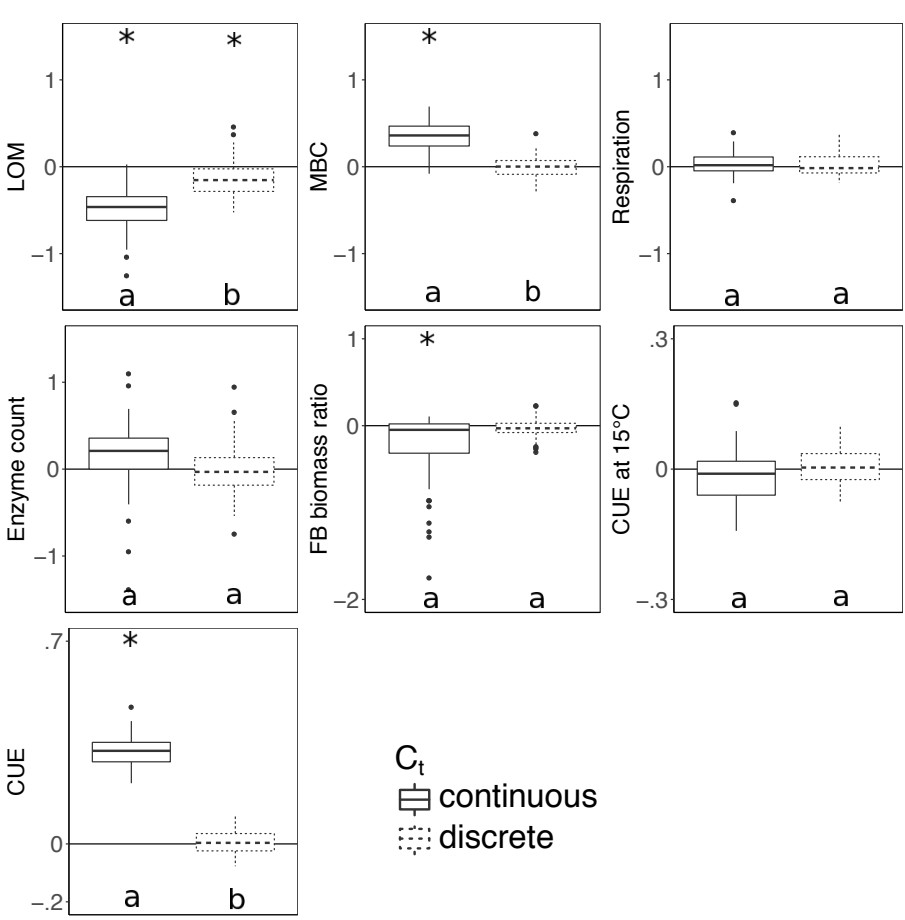

**Figure 3.** Effect of warming 5°C on C stocks and flows in simulations, reported as the natural log of the ratio of values in heated compared to control conditions. CUE temperature response either varied between taxa ("heterogeneous") or took on the fixed cross-sample mean temperature response of zero. Values above the zero line indicate warming increased the value, and values below indicate a decrease with warming. Boxplots denote 1st to 3rd quartiles with the median. Asterisks denote significant warming effect based on a paired Wilcoxon test at Bonferoni-corrected $P < 0.0001$. Letters denote warmed scenarios which are significantly different from one-another by the same criteria.

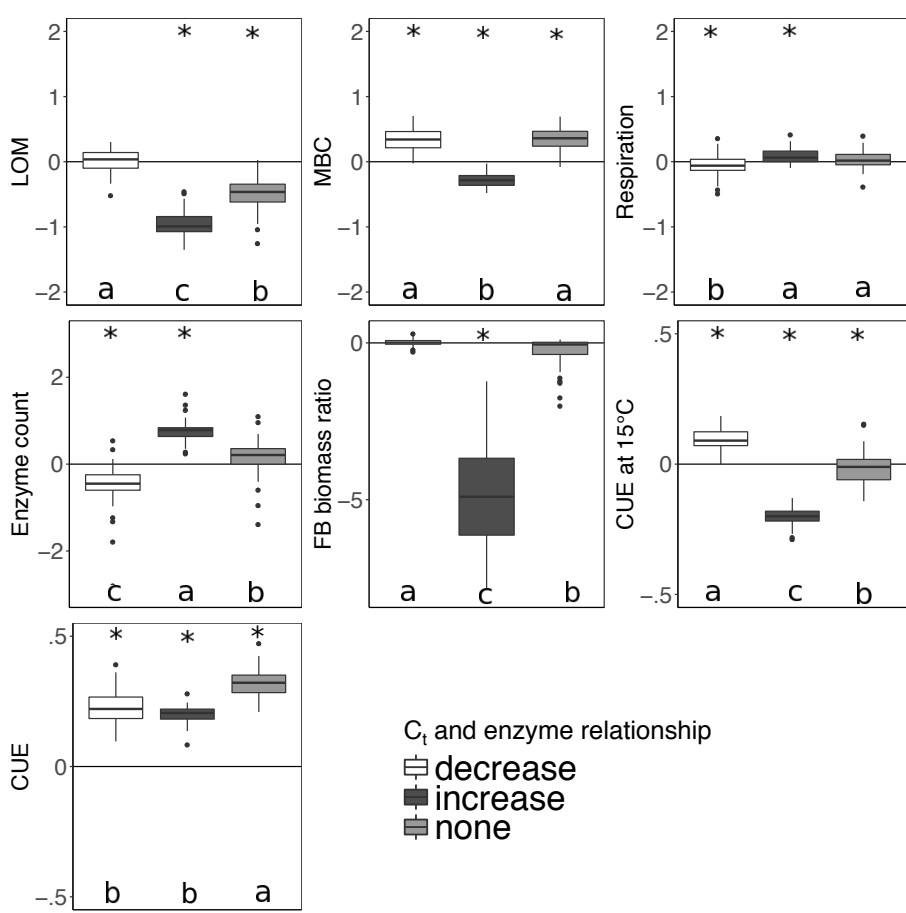

**Figure 4.** Effect of warming 5°C on C stocks and flows in simulations, reported as the natural log of the ratio of values in heated compared to control conditions. CUE temperature response was forced to increase, decrease, or remain independent of the number of enzymes a taxon could produce. Values above the zero line indicate warming increased the value, and values below indicate a decrease with warming. Boxplots denote 1st to 3rd quartiles with the median. Asterisks denote significant warming effect at $P < 0.0001$ after correcting for multiple comparisons using the Bonferoni method. Letters denote warmed scenarios which are significantly different from one-another by the same criteria.

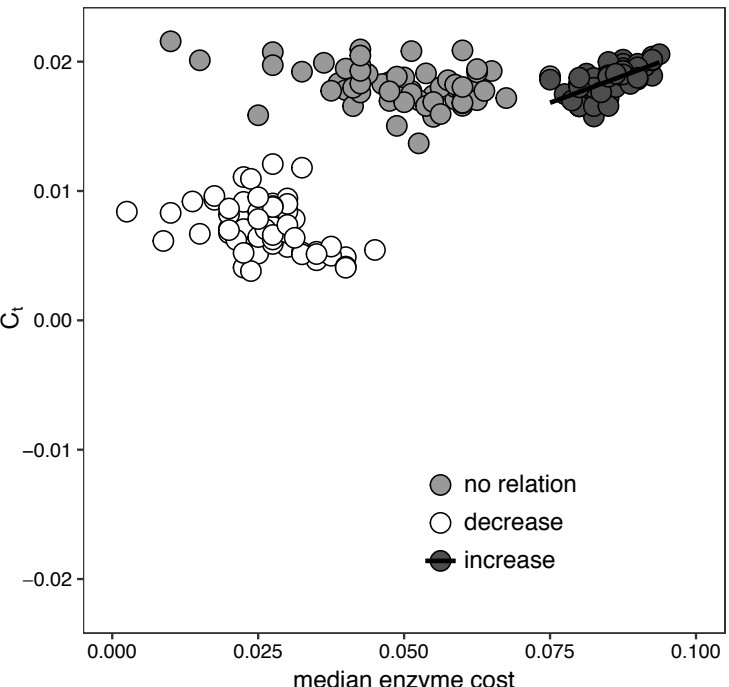

**Figure 5.** Relationship between the enzyme cost and temperature sensitivity of surviving taxa when the relationship between $C_t$ and enzyme count is set to be positive (increase), negative (decrease), or non-existent. Each point represents the value for the median across all surviving taxa for one of 59 starting communities.

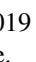



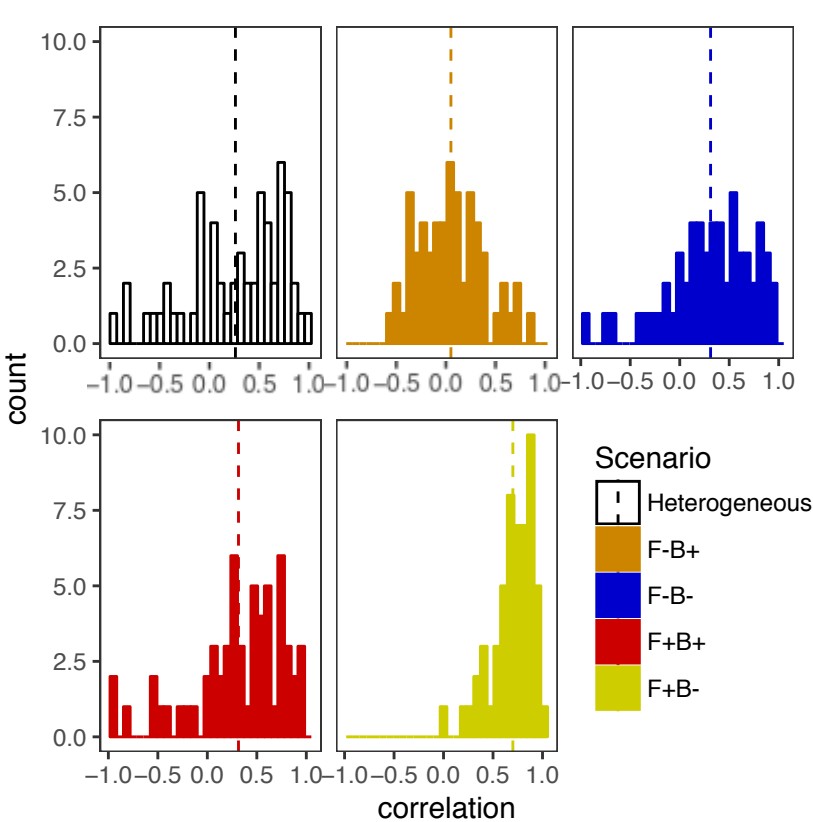

**Figure 6.** Frequency histograms of Pearson correlation coefficients between the number of enzymes taxa which survive to the end of the 6000 day run and their $C_t$. Each panel corresponds to a different scenario shown elsewhere in the manuscript, with the "counts" in each bar corresponding to a single starting seed (community) under that scenario. Vertical lines denote the mean correlation coefficient for the scenario.


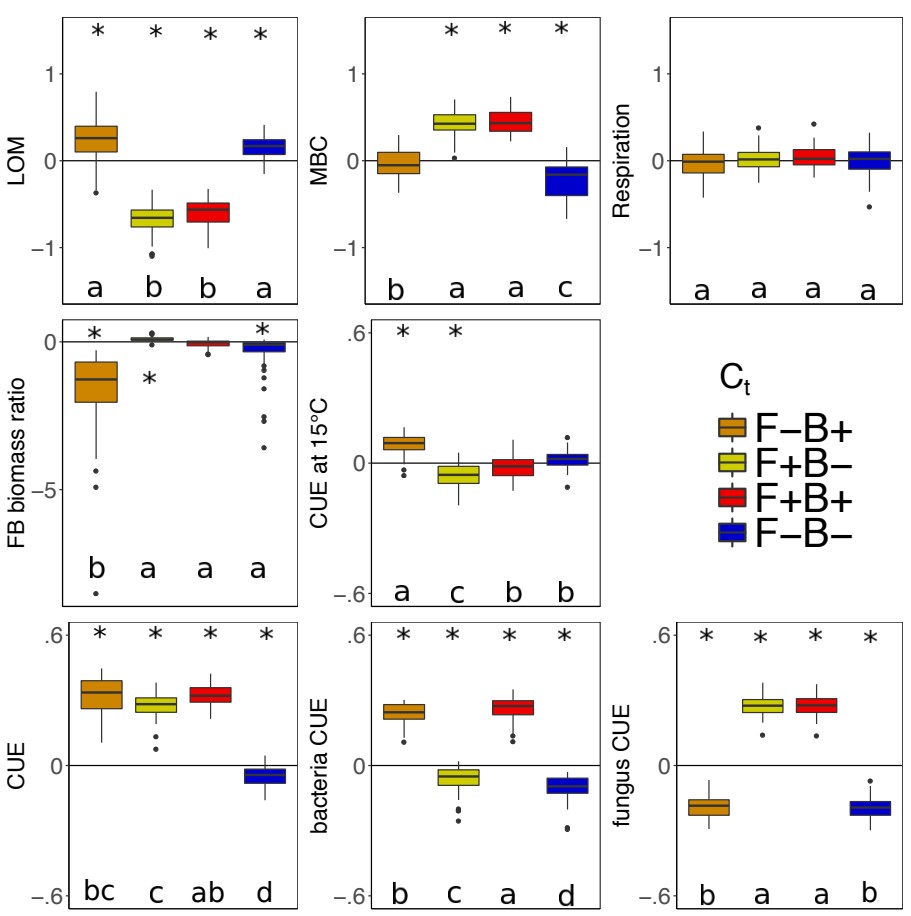

**Figure 7.** Effect of 5°C of warming on components of the litter C cycle when bacteria and fungi show dissimilar (F-B+ (orange), F+B-(yellow)) or similar (F+B+ (red), F-B- (blue)) CUE responses to temperature. Values above the zero line indicate warming increased the value (log ratio positive), and values below indicate a decrease with warming. Boxplots denote 1st to 3rd quartiles with the median. Asterisks denote significant warming effect at $P < 0.0001$ after correcting for multiple comparisons using the Bonferoni method. Letters denote warmed scenarios which are significantly different from one-another by the same criteria.





**Table 1.** CUE-related model parameters mentioned in this paper. The complete set of parameters can be found in the params.txt file in the supplement.

| Parameter | Value | Units | Description | Reference |
|-----------|-------|-------|-------------|-----------|
| $C_r$ | 0.38 | dimensionless | CUE at 15°C for a taxon with no transporters or enzymes | this paper |
| $C_e$ | -0.0025 | enzyme$^{-1}$ | change in CUE per extracellular enzyme gene | Allison, 2014 |
| $C_u$ | -0.0071 | transporter$^{-1}$ | change in CUE per transporter gene | Allison, 2014 |
| $C_t$ | -0.022 to 0.022 | °C$^{-1}$ | change in CUE per degree change in temperature from 15°C | this paper |



**Table 2.** Median (or median–standardized interquartile range (IQRm)) output values for various DEMENT model runs, marked according to warming effect (+/-) and model structure effects (letters) determined using Bonferoni-corrected post-hoc tests following linear mixed effect models. Symbols: "+" warming increased value; "-" warming decreased value). Letters: differences between warmed scenarios. All heated scenarios were compared to values in the "control" column to determine the warming effect. Only values within boxes defined by vertical lines needed to be excluded for failing to constrain litter accumulation in the two boxes. NA indicates that fungi died out completely in many instances (i.e. median fungal biomass of zero), so the parameter output could not be determined. The number in brackets denotes the median excluding the scenarios where all the fungi died out.

| Fig.1 scenario | NON-HEATED | CUE heterogeneity | | CUE-Enzyme relationship | | Fungi and bacteria differ | | | |
|---|---|---|---|---|---|---|---|---|---|
| | | A | B | C | D | E | F | G | H |
| $C_t$ (°$C^{-1}$) | NA | 0 | -0.022 to 0.022 | -0.022 to 0.022 | -0.022 to 0.022 | -0.022 to 0 (F) / 0 to 0.022 (B) | -0.022 to 0 (B) / 0 to 0.022 (F) | 0 to 0.022 | -0.022 to 0 |
| enz_cost/ $C_t$ relationship | NA | ind | ind | pos | neg | ind | ind | ind | ind |
| LOM IQRm | 0.2 | 0.14 | 0.28 | 0.33 | 0.15 | 0.39 | 0.29 | 0.23 | 0.3 |
| MBC IQRm | 0.16 | 0.16 | 0.25 | 0.02 | 0.24 | 0.14 | 0.16 | 0.15 | 0.19 |
| Surviving taxa | 9.5 | 11a | 8b | 13a+ | 7b- | 12a | 5c- | 9b | 8b- |
| Enzyme count | 16 | 16b | 20.5b | 34.5a+ | 10.8c- | 17.5a | 20.5a+ | 19a | 18a |
| Shannon's H | 1.81 | 2.01a | 1.58b- | 2.11a+ | 1.31b- | 1.93a | 1.21c- | 1.61a | 1.64a- |
| MBC (mg cm$^{-3}$) | 20.9 | 21.0b | 30.5a+ | 15.6c- | 29.3a+ | 20.3b | 33.0a+ | 32.3a+ | 17.2c- |
| LOM (mg cm$^{-3}$) | 583.2 | 496.0b- | 361.5c- | 220.1d- | 603.2a | 746.0a+ | 300b- | 334.6b- | 706.5a+ |
| CUE at 15°C | 0.23 | 0.23b | 0.23b | 0.19c- | 0.25a+ | 0.25a+ | 0.22c- | 0.23b | 0.23b |
| CUE at 20°C | NA | 0.23c | 0.32a+ | 0.28b+ | 0.29b+ | 0.32bc+ | 0.31c+ | 0.32ab+ | 0.22d- |
| FB biomass ratio | 0.86 | 0.84b | 0.81b- | 0.00c- | 0.88a | 0.12b- | 0.96a+ | 0.85a | 0.73a- |
| Respiration (mg cm$^{-3}$ day$^{-1}$) | 0.93 | 0.94b | 0.96ab | 0.99a+ | 0.88c- | 0.91a | 0.94a | 0.96a | 0.94a |
| FB richness ratio | 0.45 | 0.48a | 0.44a | 0.00a | 0.50a | 0.14c- | 0.67a+ | 0.44b | 0.40b |
| CUE bacteria | 0.27 | 0.27b | 0.26c- | 0.18d- | 0.36a+ | 0.35b+ | 0.26c- | 0.36a+ | 0.25d- |
| CUE fungus | 0.22 | 0.23c | 0.31a+ | NA(0.28)b+ | 0.28b+ | NA(0.18)b- | 0.30a+ | 0.29a+ | 0.18b- |