# Peer review of "Metabolic tradeoffs and heterogeneity in microbial responses to temperature determine the fate of litter carbon in simulations of a warmer world"

_Biogeosciences, 2019_

## Referee Comment (RC1) · Anonymous Referee #1 · 16 Aug 2019

The manuscript by Pold et al. explores the temperature sensitivity of microbial carbon use efficiency through the use of a microbial explicit decomposition model. The authors have evaluated the effects of manipulating the plasticity of CUE and enzyme production for all microbial groups or for fungi and bacteria only. The topic of the paper is highly interesting and using a modelling approach allowed for separating different effects at various scenarios. However, the manuscript seems not to be as polished as it should be for me to be able to provide an in depth review at this time. There are some aspects of the manuscript, that can be easily dealt with but make it at the moment difficult to read and to follow the story. Especially when the authors refer to Fig.1, which could be a great roadmap to the different scenarios tested, there are almost always mix

ups with which letter represents which scenario. This already starts with the figure caption. The figure itself also looks kind of distorted, at least in the pdf that I got. Lines that should be straight are not and dots are not round. In Figure 3 the legend suddenly brings up the categories 'continuous' and 'discrete', which have not been mentioned before and are not explained in the figure caption. In figure 4, the order of categories is 'decrease', 'increase', and 'none', while in the corresponding section in the manuscript it is different. Also in figure 4 there is a graph with an axis title that says 'CUE at 15°C' and one with just 'CUE' what is the difference? The reference list includes duplicates. A lot of the data that is mentioned in the text is not shown in any graphs or tables or wrongly referenced to e.g. lines 204-207. All these little things that are not grave individually make it difficult to read the paper and follow the story and make it hard for me to provide more detailed comments on the underlying science at the moment. I am however very interested in doing this, once the authors provide a more polished version of the manuscript. At this point I only have a few more general comments. 1. The title as well as the abstract are quite catchy but I think are overselling what the results of the paper show. Especially since the authors themselves state that their findings partially contradict findings from other modelling studies and empirical evidence is lacking. While the arising new concept and the new hypotheses that come with it are interesting they should be treated as such and the authors should tone down their wording, especially in the title, abstract and in the sub-headline in line 159. 2. The forth hypothesis is not very well introduced before

---

## Referee Comment (RC2) · Bin Wang (Referee) · 16 Aug 2019

This study targeted the overarching question of how CUE variability across taxa mediates microbial community in decomposing litter under increasing temperature. This question reflects a broad issue in microbial system modelling in particular and experimental studies in general, that is, an either intentional or unavoidable ignorance of CUE variation among individuals in microbial community.

To address the question, Pold et al. modified the DEMENT, a spatially explicit trait- and individual-based microbial modelling framework, and explored a series of scenarios of variability of temperature sensitivity of CUE across taxa ($C_t$). These scenarios, constrained by observed CUE variation, account for functional group (bacteria and fungi) and taxon-specific enzyme production variability. However, a few spots in the methods section could be further clarified. First, it would be better to provide an explicit equation showing how CUE is calculated in DEMENT:

CUE = Ci + (T-15)*Ct, where Ci is the intrinsic CUE, which is calculated as a function of baseline CUE and numbers of enzyme and transporter. This equation, though simple, would first make the writing much easier to organize and follow with regard to what specific changes have been modified in this study. This equation could be listed around line 80 on page 4. In addition, the first sentence in section 2.3 about Running DEMENT could be moved above to the very beginning of section 2.2 to introduce the modifications. Also, it might be better to include the DEMENT GitHub repository URL.

The results of Pold et al. clearly indicate the role of CUE variability in regulating the fate of litter C in response to temperature (and likely moisture). Although I overall agree with all of the results and discussions, there is one spot in the very beginning of the results section was particularly not clear to me: in the paragraph around line 130, first, to my understanding references to Figure. 1A and Fig.1B should be reversed. As regards the homogeneous scenario, the authors stated "all taxa had an identical temperature sensitivity that was equivalent to the cross-taxon mean (0oC-1). . .". I am not sure what exactly the constant Cr value is across taxon, 0, some value positive or negative, as illustrated in Fig.1 A, Ai, Aii? Also, throughout the discussion and results, the terms used to describe heterogeneity in the specific parameter Cr and homogeneous community are not easy to follow, and are sometimes confused with descriptions of other microbial models that are not microbial explicit. I recommend being explicit and consistent throughout the ms when using heterogeneity and homogeneity to describe the variation in Cr and across-taxon variability, as well as different models.

Overall, I believe this work sheds light on bench work and in particular modelling efforts in terms of great needs in better dealing with microbial system complexity. More specifically, I see this piece as a very helpful exercise facilitating the further development of

DEMENT and other microbial explicit models .

A final note is regarding the definition of CUE. CUE in theory is an emergent property. However, CUE in DEMENT and other models, considering a modelling tradeoff of complexity vs. efficiency, is still more like a prescribed parameter. This means current modelling framework largely cannot address causes underpinning CUE variability. Therefore, if Pold et al. could acknowledge the difference between CUE in this study and emergent CUE somewhere in the manuscript, that would make this paper even more informative, though it already is. Combined, from the novelty and implication of question addressed through soundness of methodology to writing I support the publication of this study after addressing the minor concerns above.

———————————————————

---

## Author Comment (AC1) · 29 Aug 2019

Thank you for your comments. Here is a line by line response to your concerns:

"Especially when the authors refer to Fig.1, which could be a great roadmap to the different scenarios tested, there are almost always mix ups with which letter represents which scenario"

We identified one mix-up in the figure caption and remedied it. Figure 1 has also been simplified and its modified version is attached here. The full caption is as follows: 'Schematic of experimental design used in this study, where CUE temperature

response (Ct) varies as a function of the number of enzymes and/or taxonomic affiliation of organisms. Graphs show the effects of having homogeneous (A) or heterogeneous (B) Ct across taxa; the effect of forcing a positive (C) or negative (D) correlation between the number of enzymes and Ct; bacteria showing a positive Ct and fungi a negative Ct (E); fungi showing an positive Ct and bacteria a negative Ct (F); or fungi and bacteria both having positive (G) or negative (H) Ct. Horizontal dashed lines indicate a Ct of zero, and clusters of points above and below this line denote when CUE tends to increase or decrease with increasing temperature. The letters F and B in the x-axis of individual graphs denote sensitivities for fungi and bacteria, respectively. Figure prepared in BioRender

"The figure itself also looks kind of distorted, at least in the pdf that I got. Lines that should be straight are not and dots are not round."

We have made a new copy of this figure using a different software.

"In Figure 3 the legend suddenly brings up the categories 'continuous' and 'discrete', which have not been mentioned before and are not explained in the figure caption."

The language has been remedied to be consistent throughout the manuscript.

"In figure 4, the order of categories is 'decrease', 'increase', and 'none', while in the corresponding section in the manuscript it is different"

The order has been changed to "increase", "decrease", "none" to keep in line with figure 1 and with the text where figure 4 is referenced.

"Also in figure 4 there is a graph with an axis title that says 'CUE at 15âŮęC' and one with just 'CUE' what is the difference?"

'CUE at 15oC' refers to the biomass-weighted CUE of the surviving community without 5C of warming (i.e. Cr + Ce + Cu). 'CUE' refers to the CUE at the temperature where the simulations took place (i.e. Cr + Ce + Cu + Ct*(temperature – 288K)). The appropriate equations have been added into "Analysis of outputs" section, and the two CUE

outputs have been renamed to be less confusing. Specifically, 'CUE at 15C' has been renamed to 'Reference temperature CUE', and "CUE" has been renamed to 'Simulation temperature CUE'.

"The reference list includes duplicates."

The duplicate Pietikäinen reference has been removed.

"A lot of the data that is mentioned in the text is not shown in any graphs or tables or wrongly referenced to e.g. lines 204-207."

We believe there is only one piece of data is not shown in a table or graph ('Intriguingly, neither litter (r=0.16, p=0.23) nor microbial biomass pool sizes (r=-0.44, P < 0.001) positively correlated with extracellular enzyme investment'). It is a post-hoc attempt to explain correlations between litter or microbial biomass carbon pools, which we feel can be adequately described by a simple statistic.

"1. The title as well as the abstract are quite catchy but I think are overselling what the results of the paper show. Especially since the authors themselves state that their findings partially contradict findings from other modelling studies and empirical evidence is lacking."

The title has been modified to 'Metabolic tradeoffs and heterogeneity in microbial responses to temperature determine the fate of litter carbon in simulations of a warmer world' to clarify that these are simulations. The final sentence of the abstract has been modified to reiterate the contingency of the results on model structure as follows: 'Together these results implicate a role for diversity of taxon-level CUE responses in driving the fate of litter C in a warmer world, which should be explored within the framework of additional model structures and empirical studies.'

"The forth hypothesis is not very well introduced before"

Sorry – this was a little hidden in the sentence, 'In other instances, CUE may be modeled as fixed within taxa, such that changes in community-level CUE with warming are

the result of shifts in the dominant group or groups of organisms present as a function of their dietary preferences and/or C:N ratio (Wieder et al. 2013, Sistla et al. 2014).' We appended an additional sentence to spell out this point more clearly: 'This phenomenon may occur, for instance, if large C-rich fungi show less-positive responses to warming than small, N-rich bacteria do (DeAngelis et al. 2015).'

**Fig. 1.** Revised figure 1. See comments for full caption.

---

## Author Comment (AC2) · 29 Aug 2019

Thank you for your thoughtful and helpful review. Here we will highlight the changes we made (no quotes) to the manuscript in response to your comments (in quotes):

"First, it would be better to provide an explicit equation showing how CUE is calculated in DEMENT: CUE = $C_i$ + (T-15)*$C_t$, where $C_i$ is the intrinsic CUE, which is calculated as a function of baseline CUE and numbers of enzyme and transporter. This equation, though simple, would first make the writing much easier to organize and follow with regard to what specific changes have been modified in this study. This equation could be listed around line 80 on page 4."

[Figure]

The suggested modification has been made.

"In addition, the first sentence in section 2.3 about Running DEMENT could be moved above to the very beginning of section 2.2 to introduce the modifications. "

Good idea. The sentence has been moved up.

"Also, it might be better to include the DEMENT GitHub repository URL. "

Thank you – the link has been fixed.

"The results of Pold et al. clearly indicate the role of CUE variability in regulating the fate of litter C in response to temperature (and likely moisture). Although I overall agree with all of the results and discussions, there is one spot in the very beginning of the results section was particularly not clear to me: in the paragraph around line 130, first, to my understanding references to Figure. 1A and Fig.1B should be reversed. "

Thank you – the references have been swapped and figure 1 has been modified.

"As regards the homogeneous scenario, the authors stated "all taxa had an identical temperature sensitivity that was equivalent to the cross-taxon mean (0oC-1). . .". I am not sure what exactly the constant Cr value is across taxon, 0, some value positive or negative, as illustrated in Fig.1 A, Ai, Aii? "

The cross-mean average in this scenario is zero in this instance. We have clarified this in the text and also moved Fig 1. Ai and Aii to a separate figure to reduce confusion.

"Also, throughout the discussion and results, the terms used to describe heterogeneity in the specific parameter Cr and homogeneous community are not easy to follow, and are sometimes confused with descriptions of other microbial models that are not microbial explicit. I recommend being explicit and consistent throughout the ms when using heterogeneity and homogeneity to describe the variation in Cr and across-taxon variability, as well as different models."

Thank. We have settled on using the phrasing of "fixed" CUE temperature response for those where it is a set value, "dynamic" for where CUE can change as a function of community composition changing with temperature without Ct being temperature sensitive in and of itself, and have restricted the terms "homogeneous" and "heterogeneous" specifically to describe differences in Ct across taxa.

The version of the manuscript with these revisions is now in the supplement here.

Please also note the supplement to this comment:
https://www.biogeosciences-discuss.net/bg-2019-269/bg-2019-269-AC2-supplement.zip

---

## Referee Comment (RC3) · Anonymous Referee #3 · 30 Aug 2019

This modeling experiment explored the implications of inter-taxon variability in CUE to emergent community CUE, taxonomic diversity, microbial biomass, and litter decay. Although framed within a climate change context, the work is relevant within any temperature-response system.

Overall, the work is interesting and timely but results and conclusions were sometimes overstated. Simulations are not the same as observations and at best provide possibilities when all underlying assumptions are accepted as adequate to the goals of the project.

Abstract: the next-to-last sentence was an important summary of the work, but it was

inverted and complex to unpack. Consider rewording and maybe dividing into two sentences.

The 2nd paragraph of the introduction was a good synopsis of the state-of-art for estimating CUE and logical argument for a more flexible approach.

The 3rd paragraph of the introduction makes a case for taxonomic variation in extracellular enzyme production with temperature that may affect growth efficiency, but it takes a couple of reads to understand this point. Consider revising to clarify. Also, the final sentence links rapid growth to more enzyme production, which seems counter-intuitive given that increased costs associated with producing more enzymes reroute resources from growth. This point needs more support.

The rationale for the third of the selected suite of simulations isn't clear. Is there some reason why bacteria and fungi might have opposite responses to temperature? This adds complexity to the work that needs justification.

Did the last 5000 of 6000 iterations approximate steady-state?

How many of the outputs were omitted due to unrealistically high litter accumulation? Was there any obvious reasons why this happened, such as too little biomass or constrained enzyme production? This might be an interesting result.

Line 114: what is "biomass-weighted CUE?

Why were richness and diversity calculated on different sets of data? they cannot be compared for insights to evenness, for example on line 138.

Line 135: I didn't understand the statement. Was the dampening effect a result of extending the microscale model across a macroscale environment? The next line suggests not but refers to the current study instead of the citation.

Line 138: minor issue—MBC is listed as 0.15 in text but 0.16 in table 2. Richness and diversity were not calculated on the same data, so this comparison is questionable.

Lines 140-144: Although the explanation for these patterns is logical, cause-and-effect relationships are problematic given the complexity of this model, non-linear relationships among variables, and variations in drivers.

Line 162: minor issue—is the word in parentheses "positive"?

Figure 1 wasn't easy to understand, especially the dots. Were they output or illustrative of some specific information? Didn't figures 1G and 1H contrast fungi and bacteria? The revisions were better, but still confusing.

Lines 182-4: The responses of kinetic parameters to temperature aren't consistent among the available studies, and most studies report only apparent activities. So, this statement is more speculative than it seems.

Paragraph around line 190: Maybe I missed it but some of this information would help the reader better understand the purpose of some of the selected simulation treatments if provided earlier in the modeling description.

Line 224-5: This sentence seems to conflate competition with CUE and enzyme costs, but this idea seems like it would be limited to differential responses of taxa rather than a generic response of all taxa to temperature. Right? In any case, the results of DEMENT are not necessarily proof of concept for the real world. Line 234: simulations show the results of a possible synergy within a modeling context, only.

Lines 246-9: Again, why were these scenarios chosen?

Line 260: An alternative sensitivity analysis targeting individual model characteristics defining differences between fungi and bacteria would have been appropriate, nor would it take thousands of simulations to explore.

Line 267-8: This would be a good place to remind readers of the differences in fungi and bacteria that might explain this result.

Line 290: Fitting between a range of -48 to +178% doesn't seem impressive, particu-

larly given some of the omissions. What about the omitted simulations?

Line 291-3: This is the clearest, simple statement of the study results, and its buried here.

Lines 301-3: Although interesting, this statement only adds another level of confusion. This paragraph isn't needed.

Fig. 2. Were the temperature responses of soil bacteria isolates on the 4 different media indistinguishable? Ditto for the soil community responses.

Fig. 3. How did continuous and discrete compare to heterogeneous or homogeneous? The revisions helped.

---

## Author Comment (AC3) · 17 Sep 2019

Thank you for your helpful and insightful comments on our manuscript. Please find the reviewer comments in quotes, followed by our responses.

"Overall, the work is interesting and timely but results and conclusions were sometimes overstated. Simulations are not the same as observations and at best provide possibilities when all underlying assumptions are accepted as adequate to the goals of the project."

Agreed. We have rephrased some of our results and conclusions to highlight the modeling nature of this study.

"Abstract: the next-to-last sentence was an important summary of the work, but it was inverted and complex to unpack. Consider rewording and maybe dividing into two sentences."

Thank you. We modified the sentence to read "Litter C loss was exacerbated by variable and elevated CUE at higher temperature, which effectively lowered costs for extracellular enzyme production"

"The 2nd paragraph of the introduction was a good synopsis of the state-of-art for estimating CUE and logical argument for a more flexible approach."

Thank you.

"The 3rd paragraph of the introduction makes a case for taxonomic variation in extracellular enzyme production with temperature that may affect growth efficiency, but it takes a couple of reads to understand this point. Consider revising to clarify."

Thank you. We moved up extracellular enzyme production in the first sentence to make it clear that is what this paragraph is really about, and then re-wrote the rest of the paragraph.

"Also, the final sentence links rapid growth to more enzyme production, which seems counter-intuitive given that increased costs associated with producing more enzymes reroute resources from growth. This point needs more support."

We see how it might be counter-intuitive because uptake is missing from this analysis. Fast-growing taxa need to take up more resources to maintain that growth. It doesn't mean they are growing efficiently. Hopefully the revised paragraph makes this clearer.

"The rationale for the third of the selected suite of simulations isn't clear. Is there some reason why bacteria and fungi might have opposite responses to temperature? This adds complexity to the work that needs justification."

We have observed that there is a greater positive range of Ct values for bacterial isolates relative to fungal isolates in our laboratory study (in review); however, we have thus far been unable to determine what biological factors underlie this difference. So in the absence of a known mechanism causing differences in Ct range between bacteria and fungi, we explored the effect of potential differences in Q10 within the constraints of DEMENT.

"Did the last 5000 of 6000 iterations approximate steady-state?"

No. We have never managed to get DEMENT to consistently attain a steady state. In some instances a quasi-steady state can be attained for one or two random seeds (starting communities) out of 20. But this comes with the cost of the litter C stabilizing at a concentration so high the leaves may as well be diamonds and/or those same seeds not attaining a near-steady state once the temperature is increased in the model. So for all practical purposes, we did not find a steady state attainable with the model. This is consistent with earlier work with DEMENT (ex. Allison 2014, Frontiers in Microbiology).

"How many of the outputs were omitted due to unrealistically high litter accumulation?"

0 to 5 per scenario. We have added a row to table 2 showing how many runs were excluded from each scenario based on this criteria.

"Was there any obvious reasons why this happened, such as too little biomass or constrained enzyme production? This might be an interesting result."

It is almost always due to taxon diversity loss. By chance, taxa capable of producing enzymes to effectively break down one of the substrate classes is lost, and the result is that litter component accumulates at or near the rate of daily litter addition.

"Line 114: what is "biomass-weighted CUE? "

'Biomass-weighted CUE' means that the CUE is calculated not as just the average for each taxon, but rather the contribution of each taxon to the final CUE calculation increases proportionate to the fraction of total microbial biomass that taxon contributes.

A definition has been added into the revised version.

"Why were richness and diversity calculated on different sets of data? they cannot be compared for insights to evenness, for example on line 138."

Great catch – thank you. This was an unintentional carry over from looking at model stability. The values have been updated in Table 2 and the text.

"Line 135: I didn't understand the statement. Was the dampening effect a result of extending the microscale model across a macroscale environment? The next line suggests not but refers to the current study instead of the citation."

Yes – Weider et al. 2015 proposes that microscale models lead to more homogeneous predictions as they are scaled up to the macroscale because any given single parameter will lead to fluctuations, but that the variation in physiology averages out when different phenotypic populations can asynchronously respond to their environment. However, we found that introducing variability in the CUE temperature response – equivalent to sampling a greater physiological parameter space (or pulling from a macroscale-type pool of phenotypes) – increases uncertainty in the response of litter decomposition to simulated warming. We have broken up this sentence to clarify the intended meaning.

"Line 138: minor issue: MBC is listed as 0.15 in text but 0.16 in table 2."

Thank you for catching the rounding error. The value has been corrected in the text.

"Richness and ЁЃ diversity were not calculated on the same data, so this comparison is questionable."

The values have been updated to represent calculations made on the same data.

"Lines 140-144: Although the explanation for these patterns is logical, cause-and-effect relationships are problematic given the complexity of this model, non-linear relationships among variables, and variations in drivers."

Correct. We have added a sentence reiterating that this is a conjectured driver.

"Line 162: minor issue is the word in parentheses "positive"? ËĞ Figure 1 wasn't easy to understand, especially the dots. Were they output or illustrative of some specific information? Didn't figures 1G and 1H contrast fungi and bacteria? The revisions were better, but still confusing. "

Each dot represents a taxon. The figure does not represent model output, but rather is intended to show the experimental design for the simulations (ie whether Ct was fixed or allowed to vary in a scenario, and whether the values possible in the simulation depended on the number of enzymes a taxon had or its status as a bacterium or fungus). Additional text has been added to the figure legend stating that each dot represents one of 10 example taxa in the simulations, although 100 taxa were actually used in the model. We have also updated the figure (attached) as there was a mistake in the representation of fungal and bacterial values in scenarios E-H.

"Lines 182-4: The responses of kinetic parameters to temperature aren't consistent among the available studies, and most studies report only apparent activities. So, this statement is more speculative than it seems."

This is an excellent point. We have modified the language to clarify that the stated relationships are assumed in the version of DEMENT we used, although they do not always represent reality.

"Paragraph around line 190: Maybe I missed it but some of this information would help the reader better understand the purpose of some of the selected simulation treatments if provided earlier in the modeling description."

Additional information has been added into the "Modifications to DEMENT" paragraph to demonstrate that some scenarios were included to maintain distributions of Ct and enzyme counts which were comparable, so that the effect of Ct could be parsed out from other drivers.

"Line 224-5: This sentence seems to conflate competition with CUE and enzyme costs, but this idea seems like it would be limited to differential responses of taxa rather than a generic response of all taxa to temperature. Right? In any case, the results of DEMENT are not necessarily proof of concept for the real world."

[reference sentence: "Increased CUE is likely needed to offset the costs of extracellular enzyme production that allow taxa to remain competitive at elevated temperatures within the framework of DEMENT; however, there is a paucity of empirical evidence regarding the hypothesized correlations between temperature sensitivity of CUE and enzyme investment in soil systems."]

"Line 234: simulations show the results of a possible synergy within a modeling context, only. "

Agreed. We have added in this clarifying statement.

"Lines 246-9: Again, why were these scenarios chosen?"

A sentence has been added in: "All four scenarios were tested in order to isolate the effect of changing taxonomic domain-Ct relationships from simplychanging Ct or taxonomic domain independently."

"Line 260: An alternative sensitivity analysis targeting individual model characteristics defining differences between fungi and bacteria would have been appropriate, nor would it take thousands of simulations to explore. "

This is correct. We did not try this, but our experience is that DEMENT becomes very unstable very fast when you tweak anything. Parsing out the drivers of the differences between bacteria and fungi would require not only changing parameters, but also the model itself, as three of the differences between the two groups are hard-coded into the model (motility, nutrient translocation, and how organisms divide). Therefore, we think the suggested sensitivity analysis is outside the scope of the present study.

"Line 267-8: This would be a good place to remind readers of the differences in fungi

and bacteria that might explain this result. "

The sentence "Nonetheless, differences in C and nutrient translocation abilities, and in cell size, stoichiometry and turnover rates still defined the two groups." was added.

"Line 290: Fitting between a range of -48 to +178% doesn't seem impressive, particularly given some of the omissions. What about the omitted simulations?"

The omitted simulations still have a respiration response to temperature within this range, although the absolute values of respiration are ∼6% lower on average than the simulations which were not excluded. The language has been changed in the text to state that the simulated changes in respiration are narrow compared to the range of observed respiration responses.

"Line 291-3: This is the clearest, simple statement of the study results, and its buried here."

Thank you. We have now reiterated it in the conclusion.

"Lines 301-3: Although interesting, this statement only adds another level of confusion. This paragraph isn't needed. "

The paragraph from L296-305 has been removed.

"Fig. 2. Were the temperature responses of soil bacteria isolates on the 4 different media indistinguishable? Ditto for the soil community responses."

Temperature response is, on average, the same for all media types for isolates. But substrate does matter for the temperature response of a given isolate. Soil community responses do differ based on substrate (ex Frey et al. 2013, glucose vs. phenol). Thus together, this indicates the potential for either species sorting or differential stimulation of communities at different temperatures. We have a manuscript in review on this.

"Fig. 3. How did continuous and discrete compare to heterogeneous or homogeneous? The revisions helped."

Thank you. We replaced all the continuous and discrete language with heterogeneous and homogeneous, respectively.

| Scenario | $C_t$-enzyme relationship | Fungal & bacterial $C_t$ distribution | Scenario | $C_t$-enzyme relationship | Fungal & bacterial $C_t$ distribution |
|---|---|---|---|---|---|

A homogeneous

B heterogeneous no enzyme relation

C increase enzyme relation

D decrease enzyme relation

E F- B+

F F+ B-

G F+ B+

H F- B-

**Fig. 1.** Revised figure 1

[Figure]

**Fig. 2.** Revised figure S1

---

## Referee Report (RR1)

Journal: Biogeosciences

Title: Metabolic tradeoffs and heterogeneity in microbial responses to temperature determine the fate of litter carbon in a warmer world

Authors: Grace Pold, Seeta A. Sistla, and Kristen M. DeAngelis

This study quantified the effects of variability in taxon-specific responses to temperature on the emergent, community-level carbon use efficiency (CUE). The community was allowed to change as a result of competitive interactions among taxa, resulting from resource acquisition accomplished by specific enzymes produced, costs of enzyme production, and variations in temperature sensitivity. Other characteristics of the system examined included community taxonomic diversity, microbial biomass, and litter decay.

The authors' responses to previous reviewer suggestions were effective. The result is a more persuasive demonstration of the likely roles that inter-taxon variability may have on key measures of decomposition. Although their primary focus is CUE, much the same argument could be made for many other microbial-based, emergent responses.

The revised manuscript made it much clearer why the particular suites of constraints were imposed on different taxa, e.g., selectively increasing or decreasing temperature responses, which was one of my earlier questions.

I think that enzymes should be included in the sum of organic matter produced by microbes; this omission reduces CUE as enzyme production increases (lines 55-56, elsewhere) although I suspect it's a modest difference, was this contribution quantified?

Minor quibble: line 137, higher LOM and MBC do not necessarily lead to higher ratios of MBC:LOM unless the higher MBC values were relatively higher than the LOM values.

Beginning on line 281, it was interesting that simulations appeared to be C-limited despite the high C:N content of litter. This is worth a short explanation, of course the baseline CUE (0.38) is relatively low and possibly more typical of SOM than litter. Any thoughts?

Lines 190-195: Excellent synopsis reiterating likely complexities underlying such responses. Although this work doesn't demonstrate that taxon-specific variation MUST be evaluated for any particular insight to decomposition, it clearly demonstrates that such resolution CAN provide mechanistic insight to community-level phenomena.

The revised Figure 1 with legend is much better.

---

## Author Response (AR2)

Grace Pold
Postdoctoral Research Scientist
Hampshire College School of Natural Science
Email: apold@umass.edu

[Figure]

Dear Dr. Weintraub,

Thank you for your feedback. We are resubmitting our manuscript titled "*Metabolic tradeoffs and heterogeneity in microbial responses to temperature determine the fate of litter carbon in simulations of a warmer world*" with the changes suggested by the reviewers. It was clear from the comments that it is still unclear to the reader that CUE is a fixed rather than emergent property of individual taxa in DEMENT. Below you can find a line-by-line response to the reviewers' comments.

Best Wishes,

Grace Pold, Seeta Sistla, and Kristen DeAngelis

**Response to Anonymous referee 1:**

*The manuscript has greatly improved since the last iteration. Thanks for considering all the reviewers'*
*comments. I think it is now easily understandable and should be of great interest to the readers of*
*Biogeosciences it definitely is to me. I only have some minor comments at this point.*
*Line 50: Remove one "an"*
Thank you.

*Line 83 and 88: I think it should be Cr in the formula in line 88. Please also check again for consistent*
*use throughout the manuscript.*
Thank you.

*Line 116-117: I couldn't find these files on the BG homepage*
These data are in OSF rather than the supplement hosted by the Biogeosciences journal website. We have
added a sentence clarifying this is where to find them.

*Line 179: I think confirming is a rather strong word here. I suggest to change this headline to: "The role*
*of Ct as an additional niche dimension"*

Thank you. This modification has been made.

*Line 323: Correct the brackets here.*
Thank you.

*Line 324: remove "should address"*
Thank you.

*Line 331: remove "to"*
Thank you.

*Figure 1 looks really nice now.*

Thank you.

**Response to Anonymous Referee #2's Comments:**

*1. In abstract, the statement on CUE definition, specifically regarding assimilation as biomass rather*
*than loss as CO2 is not accurate. not only CO2 but enzymes is not accounted for in CUE.*

We have amended the abstract to reflect that enzyme production decreases CUE.

*2. In methods section, the equation of CUE is calculation should correct the first term Ci to Cr.*

Thank you. The suggested correction has been made.

*3. In the conlusion part, the atuhros stated that 'DEMENT favors the prediction that litter will to become a net atmospheric C source in a warmer world', which i also believe not accurate and spectulative. we simply do not know the balance between inputs into and outputs from the litter pool, and the litter dynamics simulated by DEMENT does not fully reflect such a balance.*

The reviewer's comment is correct. Our intention with this statement was to clarify that our research does not definitively indicate that warming will cause leaf litter to respire more, but that our results support other empirical and modeling studies which indicate that this is a possibility. We have added in additional text (L333) to clarify this.

**Response to anonymous referee #3:**

*I think that enzymes should be included in the sum of organic matter produced by microbes; this omission reduces CUE as enzyme production increases (lines 55-56, elsewhere) although I suspect it's a modest difference, was this contribution quantified?*

We think there are two ideas being addressed by the reviewer's comment. The first is that extracellular enzyme C should be considered part of microbial biomass, and the second is that our estimates of CUE are too low at high rates of enzyme production because enzyme C should be included as a part of microbial biomass but isn't. The first idea is debatable on both technical and theoretical grounds. On technical grounds, very few empirical measurements of CUE attribute extracellular enzyme carbon to cells because doing so requires a number of very specific assumptions to be made about the fate of carbon in soil (such as in the method where CUE is calculated as the ratio of glucose-C respired to (glucose C added minus glucose C extracted at the end)). On theoretical grounds, enzymes are modeled as distinct from biomass in the model once produced; although they stay in the grid cell where they are produced until they "die", the rate of enzyme turnover/ "death" is independent of microbial biomass. To address the second point, the CUE numbers we present include the respiration costs of producing enzymes, but the C which enters into the enzymes themselves is not accounted for. That C is, however, subtracted from biomass carbon in keeping with the arguments to the first idea addressed above. Therefore, the omission of enzymes from the microbial biomass pool does not cause underestimation of CUE in the model because CUE is not calculated on actual MBC produced. Only a very small fraction of the carbon taken up (<1%) is parameterized to become enzyme C, so this direct cost to microbial biomass C is minimal compared to the respiration costs. Please let us know if we mis-understood the comment though.

*Minor quibble: line 137, higher LOM and MBC do not necessarily lead to higher ratios of MBC:LOM unless the higher MBC values were relatively higher than the LOM values.*

Thank you. The language has been modified to "LOM and MBC content were both generally higher than observed in environmental samples, AND CORRESPONDED TO MBC:LOM ratios at the high end of ranges observed in the field"

*Beginning on line 281, it was interesting that simulations appeared to be C-limited despite the high C:N content of litter. This is worth a short explanation, of course the baseline CUE (0.38) is relatively low and possibly more typical of SOM than litter. Any thoughts?*

CUE is not calculated based on actual MBC produced per actual $CO_2$ respired, but rather is used to determine how much of the C taken up can be allocated to biomass, enzymes, and transporters (ie it is a factor which acts instantaneously at the moment carbon is taken up). As such, CUE is unaffected by the stoichiometry of the substrate. The apparent C limitation is therefore driven by the relatively low CUE, not litter stoichiometry. We have added in a note about this on L283.

[revised manuscript text omitted]